# Survey of the Transcription Factor Responses of Mouse Lung Alveolar Macrophages to *Pneumocystis murina*

**DOI:** 10.3390/pathogens10050569

**Published:** 2021-05-08

**Authors:** Theodore J. Kottom, Kyle Schaefbauer, Eva M. Carmona, Andrew H. Limper

**Affiliations:** Thoracic Diseases Research Unit, Departments of Medicine and Biochemistry, Mayo Clinic College of Medicine, Rochester, MN 55905, USA; schaefbauer.kyle@mayo.edu (K.S.); carmona.eva@mayo.edu (E.M.C.); limper.andrew@mayo.edu (A.H.L.)

**Keywords:** *Pneumocystis*, transcription factor, HIF-1A, PPAR-γ

## Abstract

*Pneumocystis jirovecii* is a fungal pathogen that can cause life-threatening infections in individuals who are immunocompromised. Acquired via inhalation, upon entering the respiratory tract, the fungi first encounter innate immune cells such as alveolar macrophages (AMs). Relatively little is known about the AM cellular responses to the organism, and particularly transcription factor (TF) profiles leading to early host responses during infection. Utilizing the Mouse Transcription Factors RT^2^ Profiler™ PCR Array, we report an initial TF survey of these macrophage and *Pneumocystis* interactions. Expression levels of a panel of mouse TFs were compared between unstimulated and *Pneumocystis murina*-stimulated AMs. Interestingly, a number of TFs previously implicated in pathogen–host cell interactions were highly up- or downregulated, including *hif1a* and *Pparg*. qPCR experiments were further conducted to verify the results of these surveyed transcripts. Furthermore, with immunoblotting, we show that HIF-1A and PPAR-γ are indeed significantly upregulated and downregulated, respectively. Lastly, and importantly, we report that in the mouse model of *Pneumocystis* pneumonia (PCP), which mimics human *Pneumocystis jirovecii* pneumonia (PJP), qPCR analysis of *Pneumocystis murina* lungs also mimic the initial TF profile analysis, suggesting an importance for these TFs in immunocompromised hosts with *Pneumocystis* pneumonia. These data demonstrate the use of TF profiling in host AMs and *Pneumocystis* organism interactions that may lead to a better understanding of the specific inflammatory responses of the host to *Pneumocystis* pneumonia and may inform novel strategies for potential therapeutics.

## 1. Introduction

*Pneumocystis jirovecii* is recognized as one the most important fungal pathogens causing pneumonia during immunosuppression. *P. jirovecii*, which causes *Pneumocystis jirovecii* pneumonia (PJP) in humans, contributes to greater than 400,000 life-threatening fungal infections per year, with mortality rates as high as 20–80% [1]. Although highly active antiretroviral therapy (HAART) has significantly reduced PJP in the human immunodeficiency virus (HIV)-infected population, in areas where medical resources are scarce, such as sub-Saharan Africa, PJP continues to be a major problem [2]. Most current data suggests that even in developed countries, cohorts of non-HIV PJP are becoming more common, and these patients have higher mortality and unfavorable outcomes in hospitals [3]. Taken together, PJP continues to be a major contributor to morbidity and mortality in the world.

Colonization of the host lung by *Pneumocystis* and initial interactions with the host are incompletely understood. One of the first myeloid cells that the organism makes contact with are alveolar macrophages (AMs). Accumulating evidence suggests that AMs provide key roles during PJP. *Pneumocystis* organisms bind through C-type lectin receptors (CLRs) on their surfaces, activating downstream proinflammatory events such as cytokine release via engagement of the CARD9 pathway. Recently, others have shown the role of specific transcription factor (TF) responses in host–fungal pathogen interactions. For example, in *Aspergillus fumigatus* pathogenesis, hypoxia-inducible factor 1 (HIF-1A) is vital for proper inflammatory signaling to prevent uncontrolled fungal growth in the lung. Likewise, the absence of HIF-1A in host lung response to *Histoplasma capsulatum* resulted in a significant decrease in mouse survival as early as 3 days post infection, thought to be a result of increased IL-10 production [4].

With the importance of AMs in *Pneumocystis* pneumonia [5,6,7,8], and the paucity of data on what the TF response is to the fungal pathogen, we sought to conduct a survey of the TF responses via a global profiling platform in AM following *Pneumocystis* interactions. These results may provide a foundation for future specific in-depth analyzes of TF pathways with importance in *Pneumocystis* pathogenesis as well as potential areas for therapeutic development.

## 2. Results

### 2.1. Pm Infection of Mouse AMs Results in a Variety of Transcription Factor Reponses

AMs are vital for regulating anti-*Pneumocystis* immunity and lung injury. Despite this, AM responses at the TF level leading to specific proinflammatory responses are poorly understood. Therefore, we sought to explore the TF response of AMs in the presence *of Pneumocystis*. Briefly, we infected wild-type (WT) mouse AMs with *Pneumocystis murina* (Pm) to determine the expression levels of 84 TFs following 18 h culture utilizing the Qiagen RT^2^-PCR TF array. These initial experiments were repeated three times on separately isolated AMs and Pm populations and total RNAs were pooled. Before the arrays were conducted, we confirmed that all three experiments displayed similar TNF-alpha responses. Indeed, the application of Pm organisms to the AMs gave significant, robust, and repeatable TNF-alpha secretion profiles as measured by ELISA (Appendix A). Furthermore, upon analysis, we found a number of TFs either down- or upregulated that have been previously implicated in *Pneumocystis* AM interactions, including previously upregulated TFs *Irf1*, and *Nfkb1*, and in the murine *Pneumocystis* pneumonia (PCP) model, we observed the downregulation of *Egr1* [9]. Although little is known about the host TF responses of AMs to *Pneumocystis*, these confirmatory results give further support to our PCR array findings. From our PCR array experiments, we determined that of the 84 TFs analyzed, 17 were upregulated between 1.5- and 10-fold, and 40 were downregulated between 1.5- and 10-fold. These total 57 TFs are listed in Figure 1A. We decided to further analyze two specific TFs from the PCR array based on their previously reported importance in AM microbial pathogen interactions, *Hif-1a* and *Pparg*, both noted with orange or blue color, respectively, in the bar graph (Figure 1A) and scatterplot (Figure 1B). From the PCR array, the fold change compared to the mock control was 5.4 for *Hif-1a* and −5.8 for *Pparg* (Table 1).

### 2.2. Verification and Discovery of New TFs Hif-1a and Pparg Involved in AM and Pneumocystis Interactions

Next, quantitative PCR (qPCR) was implemented to confirm the PCR array results. Indeed, upon qPCR analysis, we further confirmed that *Hif-1a* was upregulated and *Pparg* downregulated in AMs upon Pm stimulation overnight (Figure 2A). In addition to changes in the transcription of both *Hif-1a* and *Pparg* in the presence of Pm, we also measured AM protein levels of both these TFs by Western blotting. Again, similar to the PCR array and qPCR data, a significant increase in HIF-1A protein levels was noted, whereas a significant decrease was noted in PPAR-γ levels when compared and quantified compared to the housekeeping B2M protein loading control (Figure 2B,C).

### 2.3. TFs Hif-1a and Pparg also Display Similar Expression Profiles during PCP

Next, to determine if these TFs were not only affected in vitro in AMs following Pm stimulation but also in vivo during *Pneumocystis* infection, we further examined the transcription levels of both of the TFs in the CD4 lymphocyte-depleted mouse model of PCP. After 10 weeks of Pm infection with this immunosuppression model, we also found that *Hif-1a* was upregulated and *Pparg* was downregulated in total lung RNA by significant amounts (Figure 3).

### 2.4. HIF-1A TF Inhibition by Specific Inhibitor PX-478 Results in Significant Reduction in TNF-Alpha Response Upon Pneumocystis Stimulation

These data suggest the possible importance of both TFs in host responses during PCP. We and others have shown the importance of a proper host immune response to *Pneumocystis* infection, particularly late in the infection [10,11,12,13,14,15]. In patients with fulminant PJP, after prophylactic treatment is initiated, patients oftentimes experience a cytokine influx thought to be due to the death of the organisms and subsequent release of proinflammatory carbohydrates from the fungal cell wall [16,17,18,19,20,21]. The use of a timed therapeutic intervention may prove beneficial at these points to prevent the detrimental effects of this uncontrolled proinflammatory response. Based on this knowledge, we preincubated mouse AMs with the specific HIF-1A inhibitor PX-478 for one hour, prior to overnight stimulation with Pm homogenate (containing exposed cell wall carbohydrates) and then analyzed the potential effects of this drug on cytokine response with TNF-alpha ELISA. Strikingly, we demonstrated that at 5 µM concentration, PX-478 can indeed inhibit TNF-alpha to a significant degree (Figure 4).

## 3. Discussion

The TF regulation of lung alveolar macrophages during PCP is largely undefined. Herein, we investigated the TF responses of mouse AMs to the fungal pathogen *Pneumocystis*. Our data show that hypoxia-inducible factor-1 (*Hif-1a*) is significantly increased, whereas peroxisome proliferator activated receptor gamma (*Pparg*) is significantly decreased, following infection with the organism.

HIF-1A is a mammalian TF that regulates metabolism, survival, and innate immune response in the low oxygen and the inflammatory setting [22]. Specifically, it has been demonstrated that in the lung, HIF-1A enhances lung inflammation in an NF-κB-dependent fashion, and promotes downstream IL-2 and TNF-alpha release [23]. Specifically, in fungal infections such as *A. fumigatus*, hypoxic conditions occur in the lung. The absence of HIF-1A resulted in decreased proinflammatory cytokine response, greater mouse mortality, and reduced fungal clearance from the lung [22]. Others have shown that in *H. capsulatum* infection, HIF-1A was upregulated in the hypoxic granulomas of the liver and that mice deficient in HIF-1A had significantly less survival compared to WT mice [4]. Interestingly, the authors suggested that spleen tyrosine kinase (Syk) activated via synergy with complement receptor 3 (CR3) and dectin-1 receptor might aid in the elevation of HIF-1A levels by infected macrophages [4]. Based on this hypothesis, and the importance of dectin-1 in PCP [24,25], future studies looking at the role of downstream Syk activation on HIF-1A levels during PCP would be important to investigate. This study also contributes to the body of knowledge on HIF-1A in fungal infections and suggests it might also be important for proper inflammatory response during PCP. Indeed, we show that the transcription levels of *Hif-1a* are significantly increased in PCP. Because of increased levels of HIF-1A in the proinflammatory host responses to fungal pathogens, early, selective, and appropriately timed therapeutic modulation of HIF-1A activity might prevent subsequent proinflammatory cascades as a result of anti-PJP therapy that results in organism death and release of highly inflammatory β-glucans [16]. Others have already shown that in rat AMs, selective HIF-1A inhibitor PX-478 can reduce HIF-1A protein levels significantly [26]. Furthermore, we have shown in this study that in vitro preincubation of selective HIF-1A inhibitor PX-478 prior to *Pneumocystis* homogenate application results in the significant reduction of TNF-alpha response.

PPAR-γ, unlike HIF-1A, is an anti-inflammatory TF that is able to dampen NF-κB-mediated cytokine production [27]. Others have shown that both in vitro [28] and under normal non-inflammatory conditions, AMs constitutively express high levels of PPAR-γ, resulting in immune tolerance in the lung [29]. Upon microbial encounter, AMs can quickly mount a proinflammatory response [30]. It has been postulated that in AMs, low levels of PPAR-γ may be beneficial for rapid immune response to microbial challenge [29]. Others have shown that in the mouse influenza model, complete absence of PPAR-γ in alveolar macrophages can lead to significant proinflammatory release [28]. Therefore, the downregulation of this TF following AM *Pneumocystis* stimulation suggests that in addition to HIF-1A activity on proinflammatory responses, yet another coordinated TF (PPAR-γ) is important for the alveolar macrophage responses to the organism and contributes to the overall magnitude of proinflammatory response during infection.

Currently, possible relationships between HIF-1A and PPAR-γ altered expression in microbial infection are unknown at this time. Future experiments in the PCP model in both/either knockout HIF-1A or PPAR-γ mouse lines or in siRNA experiments in isolated AMs stimulated with viable *Pneumocystis* organisms may provide important evidence for these questions.

The data presented in this survey of TFs important for *Pneumocystis* interactions with alveolar macrophages provide new cellular pathways that have not been previously characterized in *Pneumocystis*–host responses. Further studies are warranted to examine the other TFs detected as being significantly altered using this PCR array survey. Such further studies promise additional valuable insights into *Pneumocystis*–host immune biology as well as potential new therapeutic targets for intervention.

## 4. Materials and Methods

### 4.1. Isolation of Pneumocystis murina

All animal experiments were conducted in accordance with the guidelines of the Mayo Institutional Animal Care and Use Committee (IACUC). *Pneumocystis pneumonia* (PCP) was induced in mice (equal numbers of C57BL/6) immunosuppressed with monoclonal GK1.5 antibody as previously described [10]. *Pneumocystis murina* (Pm) organisms were derived from the American Type Culture Collection. Pm was propagated for a period of 10 weeks in immunosuppressed mice, as we previously reported [10]. Whole populations of Pm containing both cyst (ascus) and trophic forms were purified from infected mouse lungs by homogenization and filtration through 10-μm filters.

### 4.2. Mouse Alveolar Macrophage Isolation

Alveolar macrophages (AMs) were isolated by bronchoalveolar lavage as described previously [28]. Mouse lungs were lavaged with sterile calcium and magnesium free DMEM supplemented with 0.5 mM EDTA. Recovered cells in the lavage fluid samples were pelleted by centrifugation at 300× *g* for 10 min at 4 °C and resuspended in RPMI 1640 medium containing 10% fetal bovine serum, with penicillin and streptomycin. AMs were counted using a hemacytometer and plated in duplicate at 2 × 10^5^/well in a 96-well plate.

### 4.3. Stimulation Assay of AMs with Pneumocystis murina

After plating the AMs for 2 h, Pm organisms were added to the AMs at a ratio of 2:1. Plates were then centrifuged at 500× *g* for 5 min to synchronize infection and placed at 37 °C/5% CO_2_ for 18 h. At 18 h, attached Pm organisms were removed with three washes of PBS, pH 7.4. The experiments were conducted three times. Total RNA was isolated from the AMs with the RNeasy^®^ Plus Mini Kit (Qiagen, Hilden, Germany) and pooled. cDNA (500 ng) was generated with the iScript™ cDNA Synthesis Kit (Bio-Rad, Hercules, CA, USA). Finally, generated cDNA samples were applied to the Mouse Transcription Factor RT^2^ Profiler PCR Array (Qiagen), analyzing a total of 84 transcription factor genes, and PCR and analysis were conducted according to the manufacturer’s protocol.

### 4.4. Confirmation of AMs Stimulation with Pm by TNF-Alpha ELISA

To verify that the three pooled experiments had similar stimulatory macrophage responses to the organisms before PCR array analysis, we analyzed the supernatants from all three experiments for mouse TNF-alpha release by ELISA (Life Technologies™, Carlsbad, CA, USA).

### 4.5. PCR Confirmation of RT^2^ Profiler PCR Array

A similar stimulation experiment to the one above was conducted to confirm the RT^2^ Profiler PCR Array experiments. Total RNA and cDNA generation was conducted as above and quantified on a PCR utilized on a CF96 Touch™ Real-Time PCR Detection System (Bio-Rad).

### 4.6. Immunoblot Analysis of HIF-1A and PPARγ

AMs were stimulated as above with Pm for 18 h. At 18 h, the cells were lysed in 100 μL RIPA lysis buffer (0.2 mM sodium orthovanadate, pH 7.4, with protease inhibitors) and insoluble material pelleted. Next, 5 μg of total protein per sample was separated by SDS-PAGE, and HIFα or PPAR-γ protein was detected by immunoblotting with the specific antibody anti-HIF-1A (Cell Signaling #36169T) or anti- PPAR-γ (Cell Signaling #2435T), respectively. Beta-2 microglobulin (B2M) was used as a protein loading control and detected with the appropriate specific antibody to B2M (Cell Signaling, #59035). Both HIF-1A and PPAR-γ protein levels were quantified using Image Studio™ Lite (version 5.2.5), normalized against total B2M protein levels.

### 4.7. Quantitative PCR (qPCR) Analysis of mRNA Levels in Pm-Infected Mice

For qPCR analysis of mRNA expression in Pm-infected mouse lungs and controls, lungs were harvested after 10 weeks. Lung tissue samples (30 mg) were homogenized with a TissueLyser LT (Qiagen) at 50 oscillations/sec for 5 min. Subsequent total RNA isolation was conducted as noted above. Total RNA (200 ng) was converted to cDNA as described above. The qPCR analysis was then conducted with the respective primer sets, and B2M (Appendix A) was utilized as a control gene to verify equal cDNA starting amounts.

### 4.8. TNF-Alpha Analysis with HIF-1A Inhibitor PX-478

To determine if HIF-1A inhibition could prevent proinflammatory TNF-alpha cytokine response, the selective HIF-1A inhibitor PX-478 (Selleckchem, Houston, TX, USA) was applied to mouse lung AMs isolated and prepared as above 1 h prior to the application of Pm homogenate at 5 μg/mL [11]. Plates were centrifuged at 500× *g* for 5 min to synchronize infection and incubated at 37 °C/5% CO_2_ for 18 h. At 18 h, supernatants were collected and analyzed for TNF-alpha secretion by ELISA.

### 4.9. Statistical Analysis

For multigroup data, initial analysis was first performed with analysis of variance (ANOVA) to determine overall different differences. If ANOVA indicated overall differences, subsequent group analysis was then performed by 2-sample unpaired Student *t* test for normally distributed variables. Evaluation of data was conducted using Prism 9 for macOS, version 9.0.2 (GraphPad, San Diego, CA, USA). Values of *p* < 0.05 were considered significant.

## Figures and Tables

**Figure 1 pathogens-10-00569-f001:**
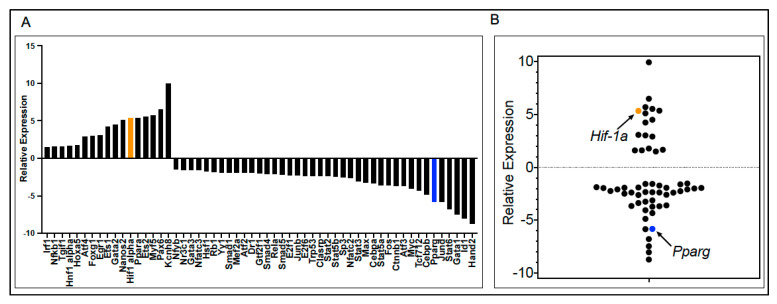
(**A**) List of up- or downregulated transcription factors (TFs) in alveolar macrophages (AMs) (isolated and pooled from at least three mice) following with *Pneumocystis murina* (Pm) infection. Duplicate wells were used and averaged. (**B**) Scatter plot of the 57 TFs. Orange bar or circle plots designate *Hif-1a* and blue color illustrates *Pparg*.

**Figure 2 pathogens-10-00569-f002:**
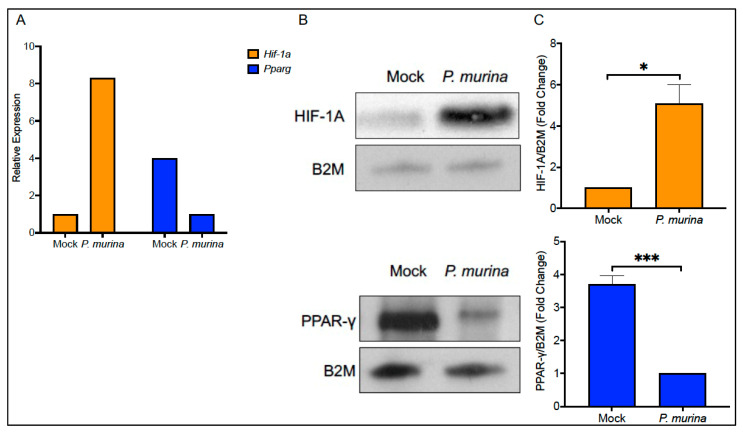
(**A**) Relative expression levels of *Hif-1a* and *Pparg* levels in isolated AMs with or without Pm infection for 18 h determined by qPCR from duplicate wells and averaged. (**B**) Western blot analysis of HIF-1α and PPAR-γ levels in AMs with or without Pm infection for 18 h. (**C**) * *p* < 0.05, *** *p* < 0.001. The bar graphs represent the relative density of two to three independent Western blot experiments as compared to the beta-2 microglobulin (B2M) loading control.

**Figure 3 pathogens-10-00569-f003:**
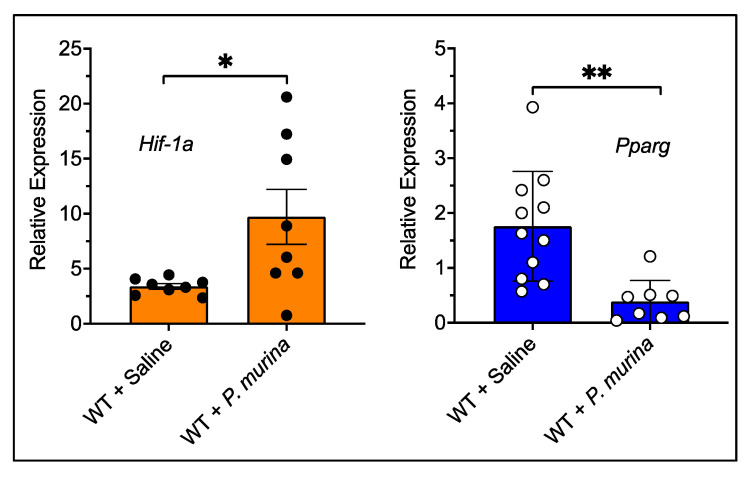
The mRNA expression level of *Hif-1a* and *Pparg* during *Pneumocystis* pneumonia (PCP). The mRNA expression levels were determined in the infected animal lungs after 10 weeks of infection. The mRNA levels were quantified by qPCR as compared to B2M mRNA levels. A total of 8–11 mice per group were analyzed. For each PCR condition, the sample was measured and averaged from duplicate wells. * *p* < 0.05, ** *p* < 0.01.

**Figure 4 pathogens-10-00569-f004:**
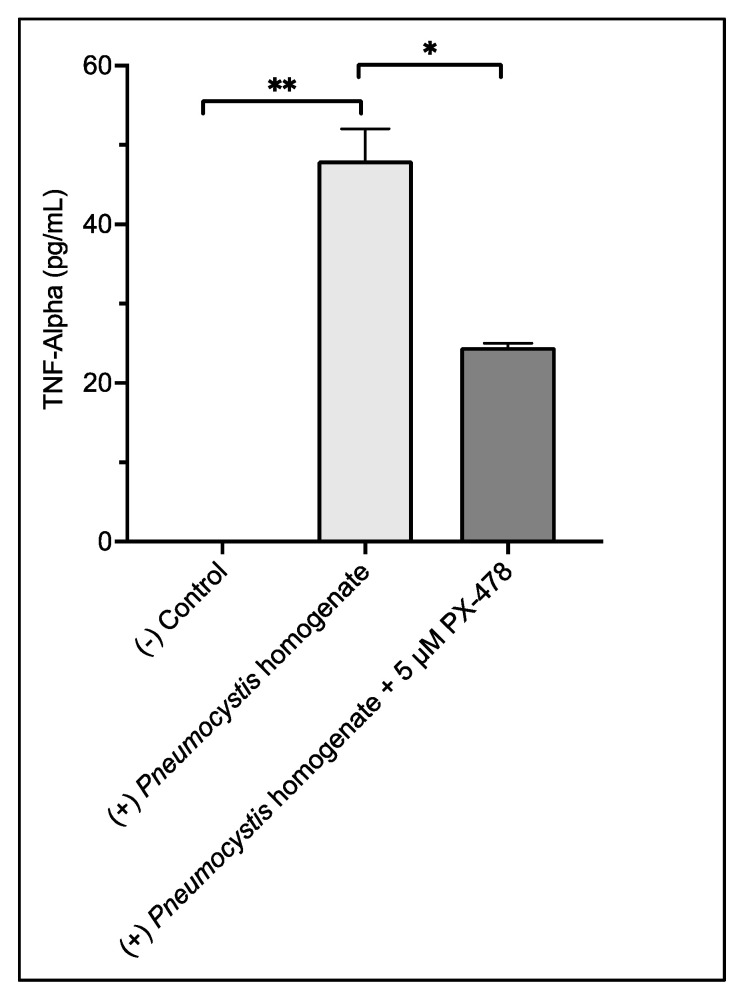
PX-478 significantly dampens AM cell production of TNF-alpha in vitro in the presence of *Pneumocystis* homogenate. The bar graph represents the results from three independent experiments with duplicate wells measured and averaged per condition. * *p* < 0.05, ** *p* < 0.01.

**Table 1 pathogens-10-00569-t001:** Denotes the RT^2^ PCR array relative expression fold change of mock versus Pm-infected AMs.

	*Hif-1a*	*Pparg*
Mock	1	1
*P. murina*	5.4	−5.8

## Data Availability

Not applicable.

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
