# Peer review of "Survey of the Transcription Factor Responses of Mouse Lung Alveolar Macrophages to Pneumocystis murina"

_pathogens, 2021, doi:10.3390/pathogens10050569_

Round 1

Reviewer 1 Report

This manuscript is a brief report examining transcription factor expression changes in cultured alveolar macrophages upon stimulation with P. murina.  The authors utilized a PCR array to identify transcription factor changes and then chose two to verify using PCR and western blot analysis.  They also verified changes in the transcription factor expression in the lungs of P. murina infected mice.  Finally, they utilized an inhibitor of HIF1α to demonstrate that it has functional significance.  Overall, the experiments are well done and make an interesting contribution to the literature surrounding the interactions between P. murina organisms and alveolar macrophages.  In figure 3 the individual symbols in the Pparg panel are not visible and should be a different shade or color.  In figure 4 it isn’t clear why the macrophages were stimulated with a homogenate of P. murina as opposed to whole organisms. 

Author Response

The authors would like to thank greatly the reviewer for their helpful comments.

The reviewer had two main comments they would like addressed:

  1. Reviewer would like in Figure #3 the individual mouse numbers a different shade color because they do not show up well against the blue background of the Pparg bar graph. As requested by the reviewer, the authors changes those individual mouse numbers to white circles with black border.
  2. Reviewer wondered why in this experiment we used a Pneumocystis homogenate instead of whole Pneumocystis organisms. As stated in the lines 118-123, we used a Pneumocystis homogenate because we wanted to emulate in vitro the potential of using a HIF1-alpha after anti-Pneumocystis treatment in which organisms are killed and release pro-inflammatory mediators such as Beta glucans that are typically hidden by the Pneumocytsis major surface glycoprotein. The outer surface glycoprotein layer often makes the viable fungal organisms (especially trophic forms) refractory in light of of the host immune pro-inflammatory response.  

Reviewer 2 Report

The current manuscript demonstrates that the coculture of Pm with AMs induces HIF1a mRNA and protein, but reduces PPARg mRNA and protein. The authors confirm that these changes are reflected in the total lung RNA of Pm infected mice. 

-The focus of the manuscript is the AM, but the authors only report expression data for whole lung tissue from the in vivo model. A direct measure of AM protein expression would strengthen their findings by showing that AM act similarly in vivo.

-Is there a relationship between altered HIF1a or PPARg expression and the AM killing capacity?

-The authors should confirm that PX478 reduces HIF1a protein in cultured AM.

-The authors should clearly state the biological sample size for each figure.

-Supplementary Figure 1 is noted in the text, but no Figure S1 was included. 

Author Response

Please see attached Word document.

Round 2

Reviewer 2 Report

The authors provide suitable answers to address the first three points raised in the initial critique. These answers and references should be concisely incorporated into the Discussion section for the benefit of the reader.

The strain and sex of the mice used should be noted in the methods section.

Author Response

POINT#1: The authors provide suitable answers to address the first three points raised in the initial critique. These answers and references should be concisely incorporated into the Discussion section for the benefit of the reader.

The authors would like to thank the reviewer for their helpful comments. Upon their recommendation, we have now incorporated the our answers and references into the discussion section.

POINT#2: The strain and sex of the mice used should be noted in the methods section.

Upon the reviewer's request we have now added the sex and strain of mice to the methods section.